# Acceptability of early childhood obesity prediction models to New Zealand families

Éadaoin M. Butler[1,2], José G. B. Derraik[1,2,3]*, Marewa Glover[4,5], Susan M. B. Morton[1,6,7], El-Shadan Tautolo[1,8], Rachael W. Taylor[1,9], Wayne S. Cutfield[1,2]*

**1** A Better Start–National Science Challenge, Auckland, New Zealand, **2** Liggins Institute, University of Auckland, Auckland, New Zealand, **3** Department of Women's and Children's Health, Uppsala University, Uppsala, Sweden, **4** School of Health Sciences, College of Health, Massey University, Auckland, New Zealand, **5** Centre of Research Excellence Indigenous Sovereignty & Smoking, Auckland, New Zealand, **6** Centre for Longitudinal Research–He Ara ki Mua, The University of Auckland, Auckland, New Zealand, **7** School of Population Health, University of Auckland, Auckland, New Zealand, **8** Centre for Pacific Health & Development Research, Auckland University of Technology, Auckland, New Zealand, **9** Department of Medicine, University of Otago, Dunedin, New Zealand

* w.cutfield@auckland.ac.nz (WSC); j.derraik@auckland.ac.nz (JGBD)

## Abstract

### Objective

While prediction models can estimate an infant's risk of developing obesity at a later point in early childhood, caregiver receptiveness to such information is largely unknown. We aimed to assess the acceptability of these models to New Zealand caregivers.

### Methods

An anonymous questionnaire was distributed online. The questionnaire consisted of multiple choice and Likert scale questions. Respondents were parents, caregivers, and grandparents of children aged ≤5 years.

### Results

1,934 questionnaires were analysed. Responses were received from caregivers of various ethnicities and levels of education. Nearly two-thirds (62.1%) of respondents would "definitely" or "probably" want to hear if their infant was at risk of early childhood obesity, although "worried" (77.0%) and "upset" (53.0%) were the most frequently anticipated responses to such information. With lower mean scores reflecting higher levels of acceptance, grandparents (mean score = 1.67) were more receptive than parents (2.10; p = 0.0002) and other caregivers (2.13; p = 0.021); males (1.83) were more receptive than females (2.11; p = 0.005); and Asian respondents (1.68) were more receptive than those of European (2.05; p = 0.003), Māori (2.11; p = 0.002), or Pacific (2.03; p = 0.042) ethnicities. There were no differences in acceptance according to socioeconomic status, levels of education, or other ethnicities.

**Data Availability Statement:** The University of Auckland Human Participant Ethics Committee approves the public sharing of the data supporting the findings of the study. They are openly available

in Figshare (https://doi.org/10.17608/k6.auckland.
9961967.v1).

**Funding:** This work was conducted for A Better
Start – National Science Challenge, which is funded
by the New Zealand Ministry of Business,
Innovation and Employment. The funders had no
role in study design, data analysis or interpretation,
decision to publish, or preparation of this
manuscript.

**Competing interests:** The authors have no
financial or non-financial conflicts of interest to
disclose that may be relevant to this work.

## Conclusions

Almost two-thirds of respondents were receptive to communication regarding their infant's risk of childhood obesity. While our results must be interpreted with some caution due to their hypothetical nature, findings suggest that if delivered in a sensitive manner to minimise caregiver distress, early childhood obesity risk prediction could be a useful tool to inform interventions to reduce childhood obesity in New Zealand.

## Introduction

An estimated 40.6 million children worldwide aged 5 years and under have overweight or obesity [1]. New Zealand is no exception, where approximately 33% of children are above a healthy weight by the time they start school [2]. High body mass index (BMI) in infancy can persist into childhood and adulthood [3], and this excess weight has adverse physical and psychological effects in both the short- [4] and long-term [5]. As long-term weight loss maintenance is difficult in children and adults, obesity prevention is preferable to treatment [6] from a public health perspective.

A number of prediction models have been developed using information available at birth (or soon after) to estimate the risk of an infant developing obesity later in childhood [7, 8]. Importantly, these models do not rely on infant weight alone, but instead employ a combination of factors to predict future obesity risk, such as maternal pre-pregnancy BMI, infant gestational age, and number of household members [8]. In addition, they have been developed for use prior to the age of 2 years, before an infant can be clinically considered overweight or obese [7]. Therefore, discussions arising from use of an early childhood obesity prediction model would be about mitigating risk of future obesity, rather than addressing issues with the infant's current weight status *per se*. However, whether parents are receptive to this information and how it might change behaviour has rarely been studied. To date, two UK-based studies have tested use of such models with parents, one as a mobile phone application and the other as part of a feasibility study [7, 8]. Little can be concluded from the feasibility study due to a poor response rate; of 226 parents invited to participate in the feasibility study, only 56 completed an assessment of their infant's obesity risk, with even fewer (n = 34) returning their 6-month follow-up questionnaire [9]. No published research exists on the efficacy or uptake of the mobile phone application [7, 8]. There is limited evidence to suggest that communication of children's genetic risk of adult obesity may influence their mothers to make healthier food choices for their child [10]. However, this study assessed mothers' food choices using virtual reality immediately after receiving the risk communication, likely introducing bias. Further, in the absence of any follow-up data, it is unknown whether this influence was lasting or had any effect on children's weight status.

Several studies have assessed parental views of receiving feedback regarding their child's weight status from researchers or school-based weight screening programmes [11–14]. However, only two UK-based study have explored parents' views of prediction models for childhood obesity; one regarding hypothetical risk communication [15] and one regarding actual risk communication [16]. Participants in the hypothetical scenario expressed a desire to hear whether their infants were at risk of obesity, despite being apprehensive of judgement from health professionals [15]. However, some parents (and even health professionals delivering the communication) in the actual scenario, rejected the risk prediction and did not consider it accurate [16]. While these studies were useful first steps into understanding parental views of

early childhood obesity prediction, they may have limited relevance for New Zealand's diverse population, where obesity rates among children and adults vary considerably according to ethnicity and socioeconomic deprivation [2, 17]. Of note, in 2015/16, 20.9% of Māori and 30.1% of Pacific 5-year-olds had obesity compared to 12.7% of Europeans [2]. For Asian children, this figure was just 8.1% [2]. The present study is the first to explore the acceptability of early childhood obesity prediction in a multi-ethnic cohort of parents, caregivers, and grandparents of children aged 5 years and under from New Zealand.

## Methods

Ethical approval was granted by the University of Auckland Human Participants Ethics Committee (#020912). The study was performed in accordance with the guidelines of the New Zealand Health Research Council and National Ethics Advisory Committee. Informed consent was electronically obtained from all participants prior to them starting the questionnaire.

### Online questionnaire

The survey questions were drafted following previous literature about parental perception, understanding of, or concern regarding current or predicted childhood obesity [11–13, 15, 18–20]. An extended questionnaire was drafted and a refined version developed with input from Māori (indigenous people of New Zealand), Pacific, and other researchers, as well as relevant early childhood organisations to ensure cultural appropriateness, ease of understanding, and relevance. The specific question used to measure acceptance of early childhood obesity prediction was:

> *"We are interested in how you as a parent or caregiver would like to be given information about your child's weight. For example, at your Well Child check in their first 6 months of life, the Well Child visitor could calculate if your baby has a greater chance of putting on too much weight by the time they start school. Would you like to know this information?"* (S1 File).

The survey was constructed and offered using an online platform (Qualtrics Labs Inc., Provo, UT, USA). Internet access is high in New Zealand, with over 90% of the population using the Internet at least once in a three-month period [21]. The survey was primarily distributed through targeted posts on social media (i.e. Facebook and Twitter) due to its ability to reach a large audience in a short period. Additional social media posts specifically targeted male caregivers, and the survey was also shared via the research team's networks. A research company (Survey Sampling International, Auckland, New Zealand) was also utilised to increase participation from Māori and Pacific respondents, and those without a university education. Due to the online nature of the survey, it was not possible to record demographic details of those who chose not to respond. Participants had to be New Zealand-based parents, caregivers, or grandparents of a child aged ≤5 years. Grandparents were included in our study as it is common for children in New Zealand to be primarily cared for by their grandparents, particularly among Māori and Pacific families [22]. However, these arrangements may be formal or informal, such that a grandparent may not have official caregiver status [22]. In addition, grandparents are the most frequent providers of informal childcare in New Zealand [23].

The questionnaire was anonymous with no identifiable information recorded. Respondents could enter a prize draw to win one of 15 supermarket or fuel vouchers, except for those recruited by the research company (rewarded with points redeemable for shopping vouchers). Data collection occurred in April–June 2018. The survey took 10 to 15 minutes to complete.

Here, we focus on questions about caregivers' perceptions of prediction of early childhood obesity (defined as obesity before a child begins school, which typically occurs at 5 years of age in New Zealand) and their acceptance of this information (S1 File). If respondents had more than one child aged ≤5 years, they were asked to focus on one particular child for the entire survey. Demographic information was collected, including respondent's age, gender, education level, caregiver status (parent, grandparent, or other caregiver), and residential district. Respondents self-reported their weight and height, and proxy-reported this information for their child.

Socioeconomic status (SES) was estimated with the New Zealand Indices of Multiple Deprivation (IMD) [24]. The IMD provides an overall measure of area deprivation based on ranked Data Zones (small geographical areas with c.712 people), but also gives individual scores for seven domains of deprivation (income, employment, crime, housing, health, education, geographical access) [24]. Respondents entered their address into the survey and an in-built algorithm calculated their IMD scores. Only IMD scores were saved, thus preserving respondent anonymity.

Ethnicity was defined using the Stats NZ hierarchical system of classification, such that all respondents were assigned to a single category [25]. Ethnicity was classified in the following order: Māori, Pacific, Asian, 'MEELA' (Middle Eastern, Latin American, African), Other, and New Zealand Europeans. Given the small numbers of respondents as 'Other' and 'MEELA', these were combined as 'Other ethnicities'.

The respondent's body mass index (BMI) was calculated; overweight was defined as BMI ≥25.0 and <30.0 kg/m$^2$, and obese as BMI ≥30.0 kg/m$^2$. Children's BMI values were converted into BMI z-scores as per World Health Organization standards [26, 27]. Please note that the child's sex was not recorded due to an error, so z-scores were based on male standards (underestimating the z-scores of girls). Childhood overweight/obesity was defined as BMI z-score ≥1.036 and obesity as ≥1.645.

## Statistical analysis

Descriptive statistics were calculated for sociodemographic characteristics. Respondent's acceptance of the prediction model information was measured using a scale ranging from "definitely yes" (score of '1') to "definitely not" (score of '5'). Group mean scores were calculated using the assigned scores, with lower scores corresponding to greater acceptance. Factors associated with acceptance of early childhood obesity were examined using a general linear model, including the following categorical predictors: sex, ethnicity (European, Māori, Pacific, and Asian), education level (complete/incomplete university qualification vs high-school or lower), caregiver type (parent, grandparent, or others), and SES (less deprived vs more deprived half). The proportions of respondents who provided their own and/or their child's height/length and weight were compared within demographic characteristics using chi-square tests. Data were analysed using SPSS v25 (IBM Corp, Armonk, USA). All tests were two-tailed, with significance level at $p < 0.05$.

## Results

Overall 2,658 potential respondents accessed the survey screening page, with 1,970 questionnaires recorded (Fig 1). 36 were subsequently excluded as based on the child's birth date provided they were aged ≥6 years, leaving 1,934 responses (Fig 1). From these, 1,731 were complete (89.5%), while the remaining 203 (10.5%) were partially complete. Among the 1,934 responses included, 61.1% were Europeans, 63.2% were aged 30–44 years (Table 1); 78.5% were mothers and 9.9% were fathers. The respondents' children were on average 2.2 years of

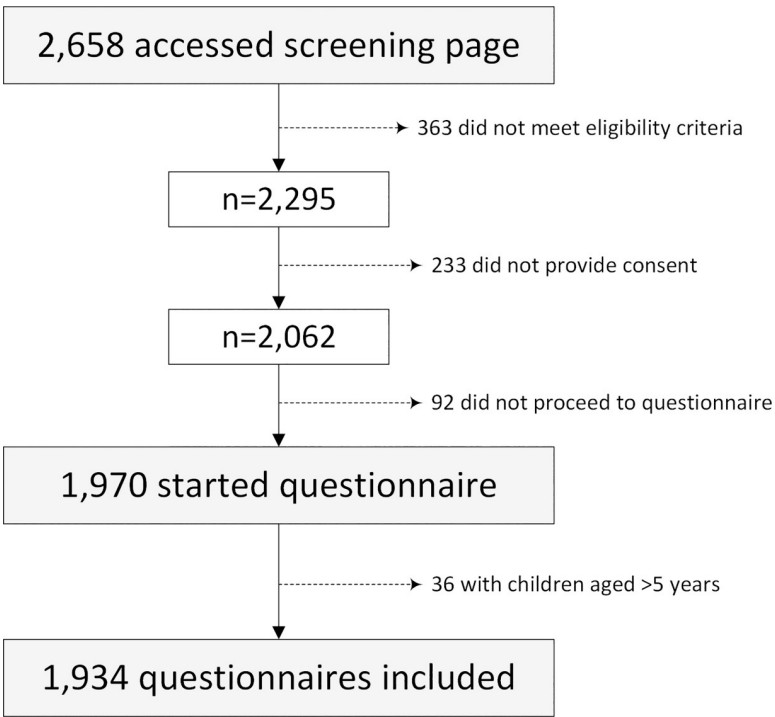

**Fig 1. Flowchart document participants' completion of online survey.**

age (SD = 1.5). Heights and weights were self-reported by 1,272 (65.9%) respondents (27.1% with obesity), and proxy-reported for 645 children (16.7% with obesity).

## Acceptability of childhood obesity prediction

When asked if they would like to know the prediction information, two-thirds (62.1%) of respondents said they would "definitely" or "probably" want to know, while 18.9% said "probably" or "definitely" not (Table 2). The interest in receiving the information according to demographic characteristics is shown in Table 2.

The results from the multivariable model are provided in S2 File. There were no differences between European, Māori, and Pacific respondents; the only distinct group were Asians (mean score 1.68), who were more accepting of the model information compared to European (2.05; p = 0.003), Māori (2.11; p = 0.002), and Pacific (2.03; p = 0.042) respondents (S2 File).

Male respondents (mean score 1.83) were more accepting than females (2.11; p = 0.005). In addition, grandparents were markedly more accepting (mean score 1.67) than parents (2.10; p = 0.0002) and other caregivers (2.13; p = 0.021) (S2 File). Respondents' acceptance of the information did not differ according to SES (1.91 vs more deprived 2.02; p = 0.09) or level of education (university 1.94 vs high-school or lower 1.99; p = 0.50) (S2 File).

## Communication of prediction information

Fig 2 shows respondents' choices for communication of the prediction information. Almost 90% (88.5%) of respondents wanted a healthcare professional to deliver the prediction information, with "knowledgeable" being the most frequently selected quality (83.0%) for this person to have. There was no single clear preference for timing of receiving the information, although the infant's transition to solid foods was selected most often (37.3%). Almost 70%

**Table 1. Demographic characteristics of questionnaire respondents.**

| | | n | % |
|---|---|---|---|
| **Overall** [1] | | 1,934 | 100 |
| **Respondent category** | Parent | 1,692 | 87.5 |
| | Grandparent | 174 | 9.0 |
| | Other caregiver | 68 | 3.5 |
| **Gender** | Male | 212 | 11.9 |
| | Female | 1,570 | 88.0 |
| | Other | 3 | 0.2 |
| **Ethnicity** | European | 1,091 | 61.1 |
| | Māori | 437 | 24.5 |
| | Pacific | 125 | 7.0 |
| | Asian | 113 | 6.3 |
| | Other ethnicities | 19 | 1.1 |
| **Born in New Zealand** | Yes | 1,404 | 78.7 |
| | No | 381 | 21.3 |
| **Education** | No qualification | 113 | 6.5 |
| | High-school qualification | 363 | 20.2 |
| | Post-school vocational qualification | 391 | 21.8 |
| | University degree [2] | 927 | 51.7 |
| **Socioeconomic status** [3] | Higher | 692 | 43.9 |
| | Lower | 883 | 56.1 |
| **Age group (years)** | 18–29 | 454 | 25.4 |
| | 30–44 | 1,129 | 63.2 |
| | 45+ | 202 | 11.3 |
| **Child age** | 0–5 months | 216 | 11.2 |
| | 6–11 months | 272 | 14.1 |
| | 1 year | 417 | 21.6 |
| | 2 years | 399 | 20.6 |
| | 3 years | 284 | 14.7 |
| | 4 years | 255 | 13.2 |
| | 5 years | 91 | 4.7 |

[1] Not all 1,934 respondents answered all questions (except for respondent category); *n* (%) for individual categories are: education (1,794; 93.0%), gender, ethnicity, birth in New Zealand, and age group (1,785; 92.3%), and socioeconomic status (1,575; 81.4%).

[2] This category includes those currently undertaking tertiary study.

[3] Socioeconomic status was estimated using the New Zealand Index of Multiple Deprivation (IMD) [23], with 'Higher' defined as all ranks 1–5 and 'Lower' as IMD overall ranks 6–10.

(69.9%) wanted to hear the information face-to-face. Respondents were concerned that receiving the information could put pressure on parents (66.7%) and the child (53.7%), while "worried" (77.0%) and "upset" (53.0%) were the most frequent anticipated emotional responses (Fig 2).

Out of 12 statements regarding various types of support to help respondents keep their baby healthy, the top four choices ranked as "very helpful" were all related to nutrition: availability of cheaper nutritious food; education about nutritious food choices; having more time to prepare healthy meals; and receiving support for breastfeeding (Fig 3).

**Table 2. Responses to the question 'Would you like to know this information?' according to gender, ethnicity, education, and socioeconomic status (SES).**

| | | Definitely yes | Probably yes | Maybe | Probably not | Definitely not |
|---|---|---|---|---|---|---|
| **Overall** | | 640 (34.3%) | 519 (27.8%) | 355 (19.0%) | 252 (13.5%) | 101 (5.4%) |
| **Gender** | Male | 81 (38.2%) | 69 (32.5%) | 41 (19.3%) | 16 (7.5%) | 5 (2.4%) |
| | Female | 529 (33.7%) | 433 (27.6%) | 303 (19.3%) | 216 (13.8%) | 89 (5.7%) |
| | Other | 1 (33.3%) | 0 | 1 (33.3%) | 1 (33.3%) | 0 |
| **Ethnicity** | European | 345 (31.6%) | 335 (30.7%) | 215 (19.7%) | 136 (12.5%) | 60 (5.5%) |
| | Māori | 153 (35.0%) | 104 (23.8%) | 89 (20.4%) | 64 (14.6%) | 27 (6.2%) |
| | Pacific | 49 (39.2%) | 28 (22.4%) | 22 (17.6%) | 22 (17.6%) | 4 (3.2%) |
| | Asian | 52 (46.0%) | 32 (28.3%) | 18 (15.9%) | 10 (8.8%) | 1 (0.9%) |
| | Other ethnicities | 12 (63.2%) | 3 (15.8%) | 1 (5.3%) | 1 (5.3%) | 2 (10.5%) |
| **Education** | No qualification | 33 (29.2%) | 31 (27.4%) | 23 (20.4%) | 15 (13.3%) | 11 (9.7%) |
| | High-school qualification | 123 (34.1%) | 94 (26.0%) | 81 (22.4%) | 46 (12.7%) | 17 (4.7%) |
| | Post-school vocational qualification | 136 (34.8%) | 105 (26.9%) | 82 (21.0%) | 48 (12.3%) | 20 (5.1%) |
| | University degree[1] | 323 (34.8%) | 273 (29.4%) | 160 (17.3%) | 125 (13.5%) | 46 (5.0%) |
| **SES** [2] | Higher | 264 (38.2%) | 187 (27.0%) | 123 (17.8%) | 90 (13.0%) | 28 (4.0%) |
| | Lower | 285 (32.3%) | 256 (29.1%) | 177 (20.1%) | 111 (12.6%) | 80 (5.9%) |
| **Respondent's age group (years)** | 18–29 | 138 (30.4%) | 135 (29.7%) | 88 (19.4%) | 64 (14.1%) | 29 (6.4%) |
| | 30–44 | 377 (33.4%) | 315 (27.9%) | 223 (19.8%) | 152 (13.5%) | 62 (5.5%) |
| | ≥45 | 96 (47.5%) | 52 (25.7%) | 34 (16.8%) | 17 (8.4%) | 3 (1.5%) |

[1]This category includes those currently undertaking tertiary study.

[2] Socioeconomic status was estimated using the New Zealand Index of Multiple Deprivation (IMD) [23], with 'Higher' defined as IMD overall ranks 1–5 and 'Lower' as IMD overall ranks 6–10.

## Weight-related concerns

More than three-quarters of respondents (77.3%) believed that they had "a lot of" or "total" control over their child's weight gain (Fig 4). In this group, 66.5% responded that they would "definitely" or "probably" want to know the model's information about their child's weight, in comparison to 46.9% of those who thought they had "some", "very little", or "no" control (Fig 4).

The vast majority of respondents (86.4%) stated that they would be "a bit" or "very" concerned if they thought their child was gaining too much weight, and two-thirds (64.9%) of them would "definitely" or "probably" want to know the prediction information (Fig 4). In contrast, this figure was 44.3% amongst the 6.5% of respondents who reported they would not be concerned at all (Fig 4).

Among respondents who provided anthropometric information for their child and stated that their child's weight gain had been fine or insufficient (n = 627), 59 (9.4%) had a child with overweight and 103 (25.8%) with obesity. Among respondents with a child with obesity who stated their child's weight gain had been fine, 92.8% also said they would be "very" or "a bit" concerned if they thought their child was gaining too much weight.

Approximately 60% (59.4%) of respondents "often" or "sometimes" had concerns about their own weight, and of these, 62.4% either "definitely" or "probably" wanted to know the prediction information on their child. The presence of obesity in respondents or their children was not associated with the respondents' levels of interest in the prediction information (Table 3). However, respondents who provided their own weight and height were slightly more receptive to the prediction information (i.e. responding "definitely yes" or "probably yes") than those who did not (64.4% vs 56.8%, respectively; p = 0.001), (Table 3). Of note,

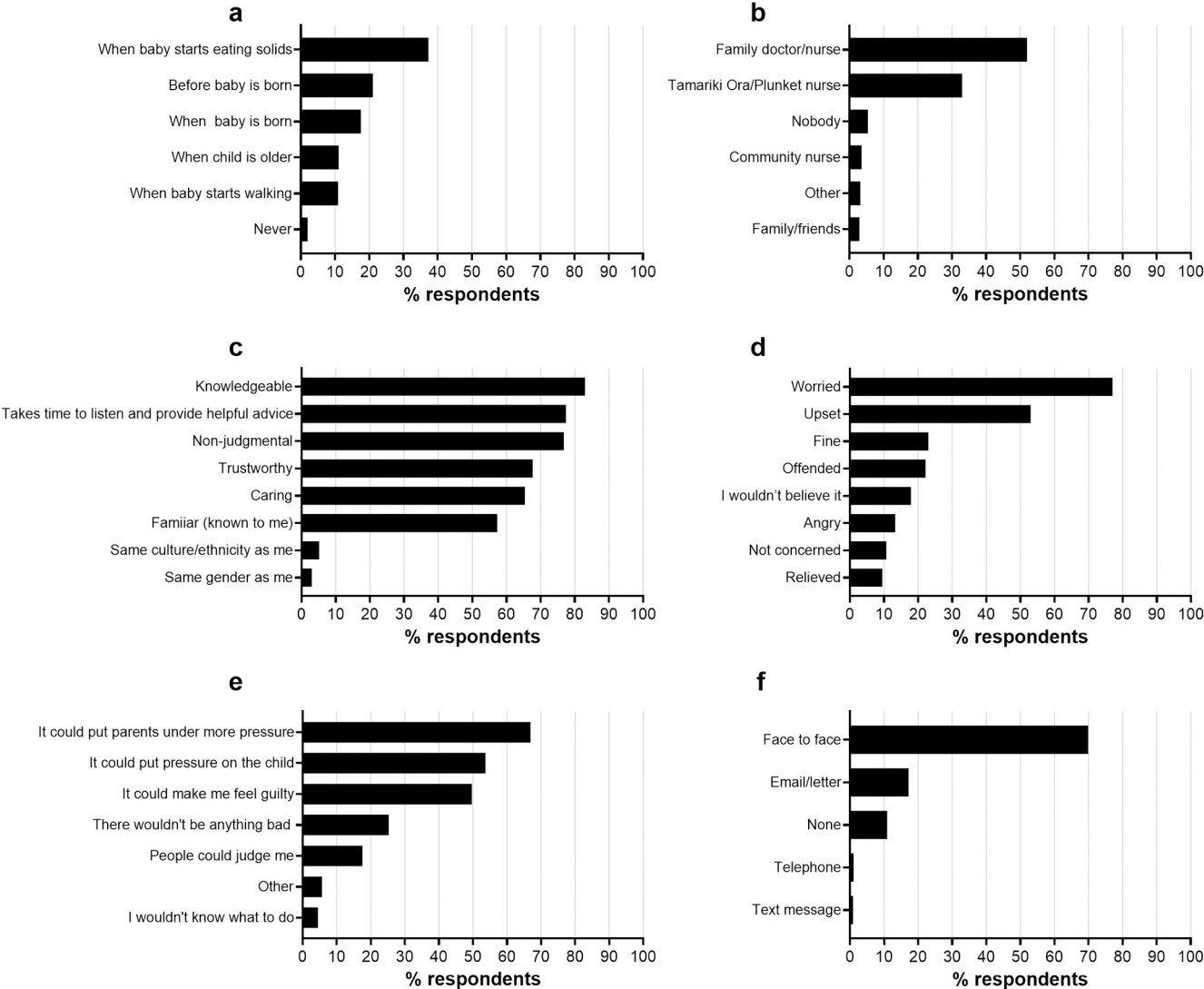

**Fig 2. Participants' responses to:** a) When do you think is the best time/stage to receive this (early childhood obesity risk prediction) information? (n = 1,818)[1] b) Who would be best to discuss this information with you, what it means, and what changes might be helpful for you and your whānau? (n = 1,867)[1] c) What important qualities should this healthcare professional have? (n = 1,820)[2] d) How would you feel if you were told your baby was at a greater risk of gaining too much weight when they are older? (n = 1,867)[2] e) What could be bad about receiving this information? (n = 1,815)[1] f) What would be your preferred way of receiving this information? (n = 1,867)[1] Footnotes: [1] Respondents could only select one answer from the options provided. [2] Respondents were able to select multiple answers from the options provided.

sociodemographic characteristics of those who provided their own and/or their child's anthropometric data were markedly different to those who did not, with this information being more frequently provided by those who were university educated, from households with lower levels of deprivation, or of European ethnicity (Table 4).

## Discussion

Using an anonymous online survey, we assessed the acceptability of early childhood obesity prediction to New Zealand-based parents, caregivers, and grandparents of children aged 5 years and under. Almost two-thirds of respondents were amenable to receiving the prediction information, with 62.1% responding that they would "probably" or "definitely" want to know.

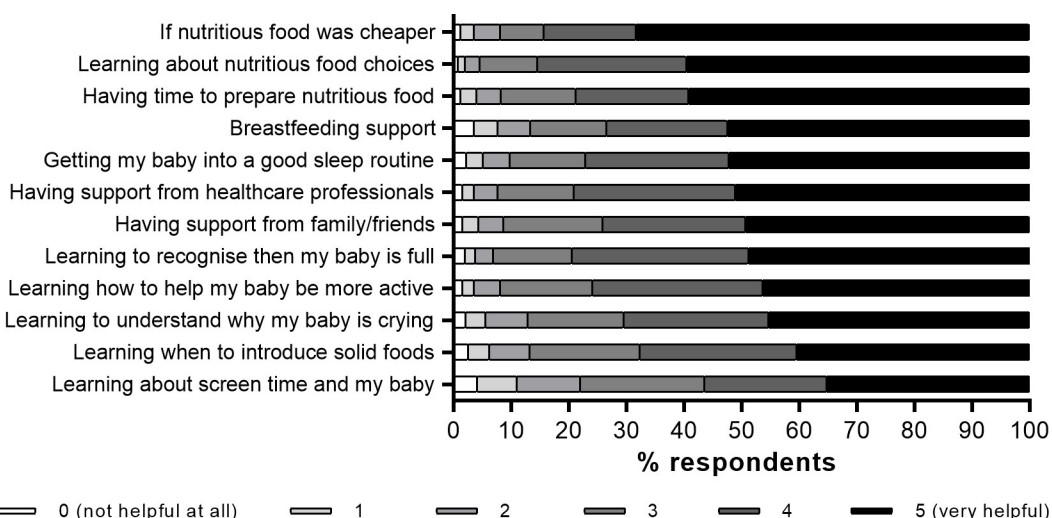

**Fig 3. Distribution of participants' ratings in response to suggested support that might help if they were told their baby was at risk of early childhood obesity (n = 1,792).**

Furthermore, there were no significant differences between responses to this question and education, or affluence, with only Asian respondents being more accepting of the prediction information than other ethnicities. More than 75% of respondents to our survey believed they had "a lot of" or "total" control over their child's weight gain. "Worried" and "upset" were the most frequently selected expected responses to being told that an infant was at risk of early childhood obesity.

Our finding that over 60% of respondents were receptive to communication regarding their baby's early childhood obesity risk supports the work of Bentley et al., who reported that respondents were generally amenable to such communication [15]. However, it is worth noting that almost 40% of respondents were ambivalent about, or not accepting of, the prediction information. Studies on parental perception of feedback regarding their child's current weight-status have shown that such feedback is considered tolerable or useful by many, but not all, parents [12, 18, 28–29]. Many parents reject the information, pointing to other indicators of their child's health as more relevant [11, 13], particularly in younger children [30]. Indeed some UK parents receiving early childhood obesity risk communication rejected this feedback, for example because they did not believe their breastfed baby could be at risk [16]. Despite overall interest in the prediction information, many respondents to our survey expected they would feel "worried" and/or "upset" if told their infant was at risk of early childhood obesity, which supports previous work showing that parents expected they would experience negative emotions in response to being told such information [15].

Our study showed that grandparents were significantly more receptive to the prediction information than parents or other caregivers. The increasing numbers of pre-school children cared for by grandparents means that the latter may play an important role in the prevention of early childhood obesity [31]. In New Zealand, Māori and Samoan grandparents responsible for feeding their young grandchildren believed that providing infants with healthy nutritional options was important, but reported significant socio-economic barriers [32]. In the UK, pre-school children from families of higher SES predominantly cared for in informal arrangements (e.g. grandparents) were more likely to be above a healthy weight at age 3 years, than children cared for in formal care settings [33, 34]. Of note, our study also showed that male respondents were more receptive to the prediction information than females. The limited available data

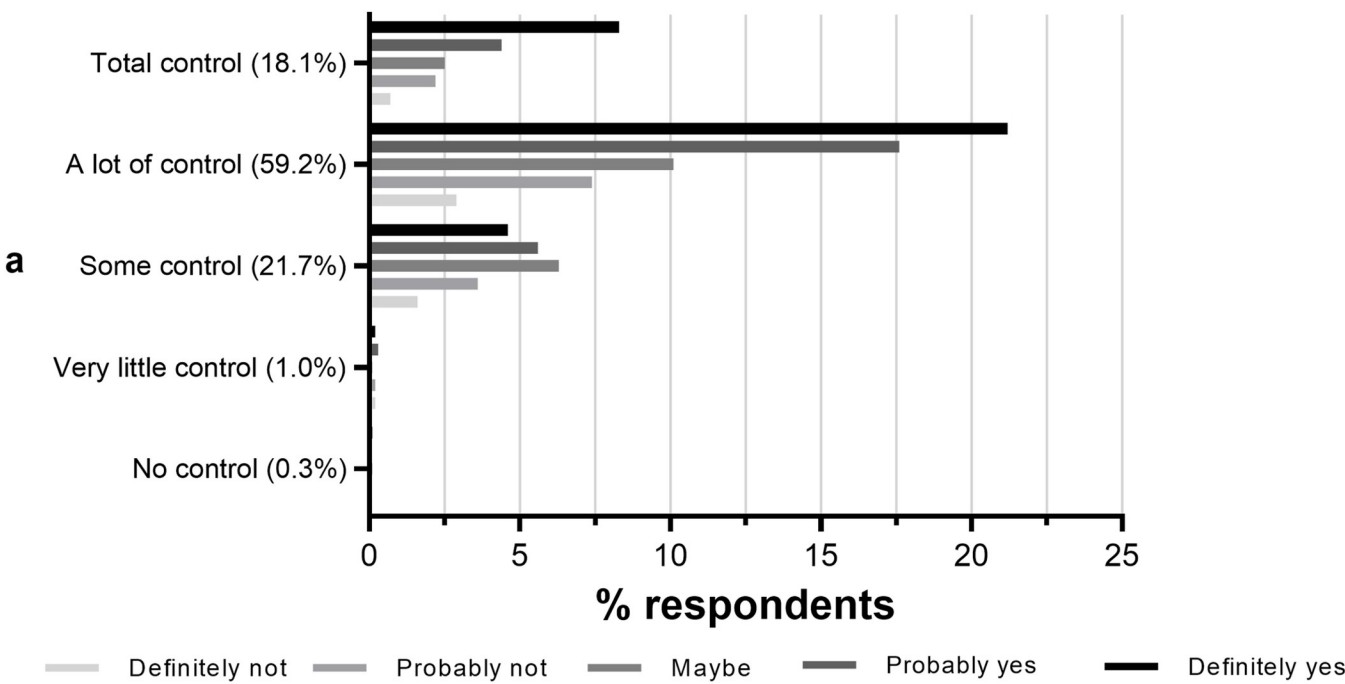

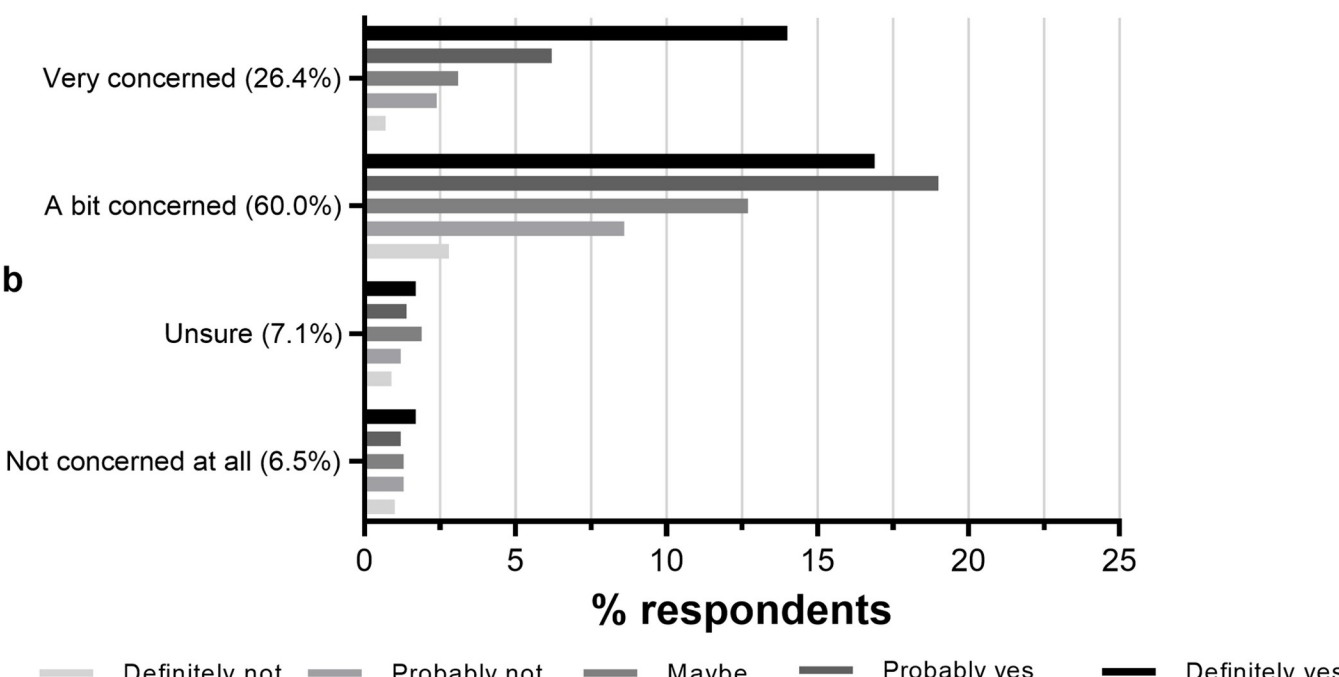

**Fig 4. Cross-tabulations of 'Would you like to know this information?' with:** a) 'How much control do you think caregivers/parents have over their child's weight?' (n = 1,867) and b) 'How concerned would you be if you thought your child was gaining too much weight?' (n = 1,867). Y axes' percentages for A and B represent overall % of responses to that question.

**Table 3. Answers to the question 'Would you like to know this information?' according to respondent weight status (n = 1,272), provision of BMI data (n = 1,867), their child's weight status (n = 645), and provision of anthropometric data on their child (n = 1,867).**

|  |  | Definitely yes | Probably yes | Maybe | Probably not | Definitely not | Total |
|---|---|---|---|---|---|---|---|
| **Respondent weight status** | Obese | 123 (37.5%) | 94 (27.2%) | 69 (20.0%) | 42 (12.2%) | 17 (4.9%) | 345 (27.1%) |
|  | Not obese | 328 (35.4%) | 276 (29.8%) | 165 (17.8%) | 115 (12.4%) | 43 (4.6%) | 927 (72.9%) |
| **Respondent provided BMI data** | Yes | 451 (35.5%) | 370 (29.1%) | 234 (18.4%) | 157 (12.3%) | 60 (4.7%) | 1,272 (68.1%) |
|  | No | 189 (31.8%) | 149 (25.0%) | 121 (20.3%) | 95 (16.0%) | 41 (6.9%) | 595 (31.9%) |
| **Child weight status** | Obese | 34 (31.5%) | 33 (30.6%) | 20 (18.5%) | 14 (13.0%) | 7 (6.5%) | 108 (16.7%) |
|  | Not obese | 185 (34.5%) | 157 (29.2%) | 89 (16.6%) | 80 (14.9%) | 26 (4.8%) | 537 (83.3%) |
| **Respondent provided child's anthropometric data** | Yes | 219 (34.0%) | 190 (29.5%) | 109 (16.9%) | 94 (14.6%) | 33 (5.1%) | 645 (34.5%) |
|  | No | 421 (34.5%) | 329 (26.9%) | 246 (20.1%) | 158 (12.9%) | 68 (5.6%) | 1,222 (65.5%) |

suggest that fathers play an important role in the development of dietary and physical activity behaviours in their children [35]. While we cannot say why our male respondents were more receptive than females, there is no doubt that paternal involvement in childhood obesity prevention should be explored further. However, it is important to consider these findings in light of the relatively small proportion of responses received from grandparents and males (9.0% and 11.9%, respectively). It is possible that our findings simply reflect highly motivated respondents, and are not reflective of the wider population.

Childhood obesity rates are inequitably distributed in New Zealand. Accordingly, we specifically targeted our recruitment to increase participation by Māori and Pacific respondents, who did not differ significantly from European respondents in their acceptance of the prediction model information. These generally high levels of interest reported by Māori and Pacific respondents seem to contradict previous findings. One study reported that although Māori

**Table 4. Sociodemographic characteristics of those who did or did not provide their own and/or their child's anthropometric data.**

|  |  | Respondent's data | | | Child's data | | |
|---|---|---|---|---|---|---|---|
|  |  | Provided | Did not provide | p-value | Provided | Did not provide | p-value |
| **n[1]** |  |  |  |  |  |  |  |
| **Gender** | Male | 162 (76.4%) | 50 (23.6%) | 0.89 | 43 (20.3%) | 169 (79.7%) | <0.001 |
|  | Female | 1,109 (70.6%) | 461 (29.4%) |  | 602 (38.3%) | 968 (61.7%) |  |
| **Ethnicity** | European | 817 (74.9%) | 274 (25.1%) | <0.001 | 467 (42.8%) | 624 (57.2%) | <0.001 |
|  | Māori | 284 (65.0%) | 153 (35.0%) |  | 99 (22.7%) | 338 (77.3%) |  |
|  | Pacific | 74 (59.2%) | 51 (40.8%) |  | 24 (19.2%) | 101 (80.8%) |  |
|  | Asian | 84 (74.3%) | 29 (25.7%) |  | 46 (40.7%) | 67 (59.3%) |  |
|  | Other ethnicities | 13 (68.4%) | 6 (31.6%) |  | 9 (47.4%) | 10 (52.6%) |  |
| **SES** | Higher | 561 (81.1%) | 131 (18.9%) | <0.001 | 319 (46.1%) | 373 (53.9%) | <0.001 |
|  | Lower | 598 (67.7%) | 285 (32.3%) |  | 290 (32.8%) | 593 (67.2%) |  |
| **Education level** | University | 708 (76.4%) | 219 (23.6%) | <0.001 | 430 (46.34%) | 497 (53.6%) | <0.001 |
|  | Less than university | 564 (65.1%) | 303 (34.9%) |  | 215 (24.8%) | 652 (75.2%) |  |
| **Respondent's age group (years)** | 18–29 | 306 (67.4%) | 148 (32.6%) | 0.001 | 149 (32.8%) | 305 (67.2%) | <0.001 |
|  | 30–44 | 838 (74.2%) | 291 (25.8%) |  | 466 (41.3%) | 663 (58.7%) |  |
|  | ≥45 | 128 (63.4%) | 74 (36.6%) |  | 30 (14.9%) | 178 (85.1%) |  |

Data are n (%).

The proportions of respondents within demographic characteristics were compared using chi-square tests.

[1] Not all 1,934 respondents answered all questions; n (%) for individual categories are: education (1,794; 93.0%), gender (1782; 92.1%), ethnicity, and age group (1,785; 92.3%), and socioeconomic status (1,575; 81.4%).

and Pacific parents believed that childhood obesity was an issue in their communities, they would not be concerned about their own child's weight gain until there were signs that it was affecting their health [36]. Similarly, research from the Pacific Islands Families study found that the majority of parents were not concerned about their young child's future weight status, although parents of children who had an unhealthy weight status were more likely to express concern [37]. It is interesting that our Māori and Pacific respondents were receptive of the prediction information, despite it being offered at a time when the infant is likely too young to have obesity-related comorbidities or be diagnosed as above a healthy weight.

Parents who believed they had "total" or "a lot of" control over their child's weight gain and those who would be concerned about excessive weight gain were more likely to want to hear the prediction information. This suggests that the perceived risk to health posed by childhood obesity, as well as the respondent's belief in their self-efficacy to tackle the issue, might predict their openness to hearing that their child was at risk of early childhood obesity. These findings are supported by theories of health behaviour such as the Theory of Planned Behaviour [38] and the Health Belief Model [39]. Interestingly, there appeared to be no differences in respondents' interest in the prediction information according to their own or their child's weight status (obese or non-obese). However, those who did not provide their own weight and height data were apparently less receptive to the prediction information. There were clear demographic differences between those who did and did not provide anthropometric data, with those who are most likely to be affected by obesity (Māori and Pacific respondents, those with less than university education, and those of lower affluence), being the least likely to respond to these questions. In the UK, women with overweight or obesity preferred larger infants, and did not express any concerns about health risks associated with childhood obesity [40]. While there did not appear to be any differences in respondent's acceptance of the prediction information according to whether or not they provided their child's weight and height in our study, it is possible that they simply did not know this information (rather than intentionally not disclosing it).

Over 90% of respondents to our survey with a child with obesity who stated their child's weight gain had been fine, also said they would be "very" or "a bit" concerned if they thought their child was gaining too much weight. This clearly shows a disconnect between respondents hypothetical and actual concern, in that although these respondents believed they would probably be concerned by their child gaining too much weight, in reality they did not report this as they failed to recognize that their child was actually above a healthy weight. This finding also lends support to the notion that childhood obesity is a problem for somebody else's family, as noted previously [41]. Therefore, it is entirely possible that respondents would have a different perception of the information if it was communicated to them in a real-life situation. Indeed, the UK studies into parental views on risk prediction of early childhood obesity showed that while parents may be accepting of this risk communication in a hypothetical scenario [15], it may be rejected in the real world [16].

Regarding delivery of the prediction information itself, the general preference for receiving the prediction information when the infant transitions to solid foods conflicts with the findings of Bentley et al. [15], who found that their participants viewed when the infant starts to walk as the appropriate developmental phase to receive such information. Furthermore, respondents also rated various types of nutritional support as the top four most useful types of assistance for their baby to be healthy. Respondents to our survey may have based their choices to these questions on the belief that any kind of weight-related intervention in young children would be primarily diet-based. On the other hand, it is interesting to note that the second most frequently selected time point for receiving the information was before the baby is even born.

The majority of respondents wanted a medical professional to discuss the prediction information with them. Just over half of respondents selected "family doctor or nurse", with another 33% choosing "Plunket or Tamariki Ora nurse" (those delivering the nationally-funded Well Child programme from birth to age 5 years). This is in line with previous work showing that parents wanted a healthcare professional to communicate with them if their infant was gaining too much weight [40]. Also, "knowledgeable" was the most frequently selected quality that respondents wanted this healthcare professional to have, so it would seem that the credibility of the communicator is very important. Lastly, respondents were concerned that the information could place additional pressure on parents and/or the child. Lack of resources, such as knowledge, time, and finances, are often cited as barriers to parental instigation of behaviours to reduce or prevent childhood obesity [12, 42]. Thus, it is important that the information would be communicated in conjunction with ongoing support to reduce this perceived pressure and ensure that any intervention would be accessible to all parents.

There are a number of limitations to our study. First, we do not know the sex of the children of respondents to our survey; there could be differences in how receptive respondents are to the prediction information according to their child's sex, as well as limiting the interpretability of the BMI z-score data. This is compounded by the self-report nature of the information, as children's BMI z-score data may be particularly susceptible to error given that even small caregiver misjudgements of weight or height/length may lead to inaccurate estimation of weight status in this young population. However, given the online nature of the study, it was not possible to physically weigh and measure children, and thus results must be interpreted in light of this limitation. In addition, we cannot ascertain the representativeness of our sample, which could affect our ability to readily extrapolate the observed differences in acceptance between groups to the general New Zealand population. Lastly, our study assessed respondents' interest in a hypothetical risk communication, that is, none of the respondents were actually told that their infant was at risk of early childhood obesity. However, it is ethically important to conduct preliminary acceptance studies (such as ours), before communicating potentially distressing information in real world scenarios. Further research could investigate responses to receiving a real early childhood obesity risk prediction, as well as assess what, if any, impact this has on the child's future weight status. Key strengths of our study include a sample size of nearly 2,000 respondents, and our specific recruitment strategy to increase participation of Māori and Pacific caregivers, as well as males and respondents with lower levels of education. Lastly, the wording of our questionnaire was reviewed by several prominent Māori and Pacific researchers in order to ensure it was culturally appropriate.

## Conclusions

Our study has shown that almost two-thirds of respondents to our survey were receptive to communication about early childhood obesity prediction. Notably, Māori, Pacific, and European respondents had similar levels of interest in being told the prediction information. This finding is of particular importance given the inequitable rates of childhood obesity experienced by Māori and Pacific children. If early childhood obesity prediction is deemed acceptable by Māori and Pacific families, it is possible that it may be used as a resource to assist with the reduction of early childhood obesity in those communities. While our results must be interpreted with some caution due to their hypothetical nature, taken together, our findings suggest that if delivered in a sensitive manner to minimise caregiver distress, early childhood obesity risk prediction could be a useful tool to inform interventions to reduce childhood obesity in New Zealand.

## Supporting information

**S1 File. Survey questions on caregivers' views of early childhood obesity prediction models.**
(DOCX)

**S2 File. Results from a general linear model examining the associations between caregiver's demographic characteristics and their level of acceptance of the obesity prediction model information.**
(DOCX)

## Acknowledgments

The authors wish to thank Dr Daniel Exeter (University of Auckland) for valuable assistance with the New Zealand Index of Multiple Deprivation.

## Author Contributions

**Conceptualization:** Éadaoin M. Butler, José G. B. Derraik, Marewa Glover, Susan M. B. Morton, El-Shadan Tautolo, Rachael W. Taylor, Wayne S. Cutfield.

**Data curation:** Éadaoin M. Butler, José G. B. Derraik.

**Formal analysis:** Éadaoin M. Butler, José G. B. Derraik.

**Funding acquisition:** José G. B. Derraik, Rachael W. Taylor, Wayne S. Cutfield.

**Investigation:** Éadaoin M. Butler, Marewa Glover, Susan M. B. Morton, El-Shadan Tautolo, Rachael W. Taylor, Wayne S. Cutfield.

**Methodology:** Éadaoin M. Butler, José G. B. Derraik, Marewa Glover, Susan M. B. Morton, El-Shadan Tautolo, Rachael W. Taylor, Wayne S. Cutfield.

**Project administration:** Éadaoin M. Butler.

**Resources:** Éadaoin M. Butler, Wayne S. Cutfield.

**Supervision:** José G. B. Derraik, Rachael W. Taylor, Wayne S. Cutfield.

**Writing – original draft:** Éadaoin M. Butler.

**Writing – review & editing:** Éadaoin M. Butler, José G. B. Derraik, Marewa Glover, Susan M. B. Morton, El-Shadan Tautolo, Rachael W. Taylor, Wayne S. Cutfield.

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
