## [Decision Letter · Decision Letter 0]

20 Aug 2019

PONE-D-19-17554

Acceptability of early childhood obesity prediction models to New Zealand families

PLOS ONE

Dear Dr Butler,

Thank you for submitting your manuscript to PLOS ONE. After careful consideration, we feel that it has merit but does not fully meet PLOS ONE’s publication criteria as it currently stands. Therefore, we invite you to submit a revised version of the manuscript that addresses the points raised during the review process.

We would appreciate receiving your revised manuscript by Oct 04 2019 11:59PM. To enhance the reproducibility of your results, we recommend that if applicable you deposit your laboratory protocols in protocols.io, where a protocol can be assigned its own identifier (DOI) such that it can be cited independently in the future. For instructions see: http://journals.plos.org/plosone/s/submission-guidelines#loc-laboratory-protocols

We look forward to receiving your revised manuscript.

Kind regards,

David Meyre

Academic Editor

PLOS ONE

Journal Requirements:

Reviewers' comments:

Reviewer's Responses to Questions

**Comments to the Author**

1. Is the manuscript technically sound, and do the data support the conclusions?

Reviewer #1: Partly

Reviewer #2: No

Reviewer #3: Yes

2. Has the statistical analysis been performed appropriately and rigorously? 

Reviewer #1: Yes

Reviewer #2: I Don't Know

Reviewer #3: Yes

3. Have the authors made all data underlying the findings in their manuscript fully available?

Reviewer #1: No

Reviewer #2: Yes

Reviewer #3: Yes

4. Is the manuscript presented in an intelligible fashion and written in standard English?

Reviewer #1: Yes

Reviewer #2: Yes

Reviewer #3: Yes

5. Review Comments to the Author

Reviewer #1: The manuscript by Dr. Butler and co-authors deals with an interesting topic and is very well organised and presented.

However I have some concerns about recruitment and results interpretation.

In particular:

- the majority of respondents are mothers, while fathers are less than 10%. Based on statistical analyses, authors conclude that male respondents are more likely to be interested in knowing their child's risk to become obese. Did the study protocol ensure that the survey would equally reach fathers and mothers? It is not at all clear from lines 105-110. In case the survey invitation was equally effective in reaching males and females, the percentage of male respondents would show that males are in general less interested in the topics than females and probably would be less receptive to the risk prediction, which is the opposite than the authors' conclusions. The fact that the 9% of male respondents poved to be more interested than females in the risk prediction would be just a result of this bias (the little number of male respondents represents a highly motivated subgroups of males);

- the authors state that caregivers are "generally accepting of receiving information on their baby risk..." based on a 62% percentage of respondents definitely or probably wanting to know. I do not agree that 62% represents a "general acceptance". Almost 40% of respondents do not fall into the accepting category and this should be addressed;

- line 425 represents a result over-interpretation.

Moreover:

The manuscript is unnecessarily long. Some paragraphs include quite expected or not so interesting information which adds very little to the main manuscript message and could be shortened or even removed tobetter focus on the main topic. For example lines 217-219 and relative paragraphs in the discussion.

Reviewer #2: The manuscript addresses an important an interesting problem, regarding parents/families acceptability of receiving predicted obesity risk information. It is well written, but needs to be more concise. The title specifically mentions acceptability of early childhood obesity prediction models, but parents have not been asked about obesity prediction at all. Neither have they addressed this question in their methods.

Hence the main problem with the manuscript is the mismatch between what the parents are asked to comment on, and what the paper purports to be about. Whilst it is entirely appropriate that parents are not asked specifically about obesity, the authors need to address the mismatch in the title of the manuscript and the question actually posed to participants.

The manuscript refers many times to ‘obesity prediction’ and risk information, but nowhere is it the specific question posed to participants articulated in the manuscript. This is a major omission because the way the initial question is framed would likely affect the acceptability of the information.

There is one question in the appendix that refers to putting on too much weight by the time they start school. If this is the question that the authors refer to as ‘prediction information’ it needs to be in the main manuscript, but putting on too much weight ( weight trajectory) is quite different to ‘early childhood obesity prediction information’ referred to in the manuscript.

Additionally there is no example of the format the information might take. For example prediction information could be in the form of probability of childhood obesity, or probability of reaching above a heathy weight by a particular age etc etc. How will the authors convert from a prediction model to ‘chance of putting too much weight by the time they start school’????.

Results

It is unclear whether multivariable models were used for all question responses. The results appear to be simple summary statistics. If the multivariable model pertains to the main question, the model should be presented.

Line 172 ‘like to know the prediction information’ – it appears they were not asked this at all.

Line 209..’want to know the model’s information about their child’s weight’ – there is no question in the appendix that asks about information from a model.

Line 227-239 – too much detail. If this is in the tables, please don’t repeat in the text.

Figure 2 This is a nice figure, but the axis scales should be consistent, at least for the two different question types.

Supplementary appendix page 2

Question beginning ‘We are interested….. calculate if your baby has a greater chance of putting on too much weight by the time they start school.’ – greater chance than what? Average? Should this have been high chance?

It is unclear whether this is the main question that has been interpreted as the prediction information

The other major comment is the discussion is much too long, and at times veers off into a discussion of previous studies or aspects not relevant to the main theme of the paper. It needs to be much more concise. I would suggest reducing it to 25% of its current length.

Minor points

Abstract -Line 30 refers to infant, whilst results refers to child

Introduction

Line 58 – preferable to whom?

Line 61 A number of prediction models – yet only two cited

Line 66 should this be overweight or obese?

Line 69 – country context of the models should be mentioned

Line 80 – and who? GP/nurse/childhood educator

Line 81 thoughts ….should be views

Line 122 and throughout- early childhood should be defined at least once. Does this mean under 5 years?

Line 142 replace BMI with BMI values

Reviewer #3: This paper is very interesting and well-written piece of work. I enjoyed reviewing it. I have a few minor comments that I feel may strengthen the paper.

6. PLOS authors have the option to publish the peer review history of their article (what does this mean?). If published, this will include your full peer review and any attached files.

Reviewer #1: No

Reviewer #2: No

Reviewer #3: Yes: Simone Annabella Tomaz

---

## [Author Response · Author response to Decision Letter 0]

10 Oct 2019

Responses to Referees' Comments

PONE-D – 19-17554 

"Acceptability of early childhood obesity prediction models to New Zealand families"

REVIEWER #1

1. The manuscript by Dr. Butler and co-authors deals with an interesting topic and is very well organised and presented.

Reply: We thank the Reviewer for their positive feedback. 

However I have some concerns about recruitment and results interpretation.

In particular:

2. - the majority of respondents are mothers, while fathers are less than 10%. Based on statistical analyses, authors conclude that male respondents are more likely to be interested in knowing their child's risk to become obese. Did the study protocol ensure that the survey would equally reach fathers and mothers? It is not at all clear from lines 105-110. In case the survey invitation was equally effective in reaching males and females, the percentage of male respondents would show that males are in general less interested in the topics than females and probably would be less receptive to the risk prediction, which is the opposite than the authors' conclusions. The fact that the 9% of male respondents proved to be more interested than females in the risk prediction would be just a result of this bias (the little number of male respondents represents a highly motivated subgroups of males);

Reply: Unfortunately, we are unable to tell if our survey was equally effective in reaching both males and females. As our study was primarily shared online via social media, we had no way of recording exposure, and thus cannot determine a response rate. We have added a sentence to the Methods section to explain this (Lines 123-124):

“Due to the online nature of the survey, it was not possible to record demographic details of those who chose not to respond.”

We acknowledge the reviewer’s point re bias and have altered the Discussion (Lines 285-288) to reflect this. We have alluded to the same regarding our significantly higher acceptance among grandparents, as we believe this caveat also applies to that result.

“However, it is important to consider these findings in light of the relatively small proportion of responses received from these grandparents and males (9.0% and 11.9%, respectively). It is possible that our findings simply reflect highly motivated respondents, and are not reflective of the wider population.”

3. - the authors state that caregivers are "generally accepting of receiving information on their baby risk..." based on a 62% percentage of respondents definitely or probably wanting to know. I do not agree that 62% represents a "general acceptance". Almost 40% of respondents do not fall into the accepting category and this should be addressed;

Reply: We acknowledge the reviewer’s point and have made a number of edits to our manuscript, removing references to “generally accepting” and adding a point about those that were not accepting to the discussion.

Abstract (Lines 48-49):

“Almost two-thirds of respondents were receptive to communication regarding their infant’s risk of childhood obesity.”

Discussion (Lines 253-255):

“Almost two-thirds of respondents were amenable to receiving the prediction information, with 62.1% responding that they would “probably” or “definitely” want to know.”

Lines 262-267:

“Our finding that over 60% of respondents were receptive to communication regarding their baby’s early childhood obesity risk, supports the work of Bentley et al., who reported that respondents were generally amenable to such communication (15). However, it is worth noting that almost 40% of respondents were ambivalent about, or not accepting of, the prediction information. Studies on parental perception of feedback regarding their child’s current weight-status have shown that such feedback is considered tolerable or useful but a many, but not all, parents (12,16,25,26).”

Conclusions (Lines 377-378):

“Our study has shown that almost two-thirds of respondents to our survey were receptive to communication about early childhood obesity prediction.”

4. -line 425 represents a result over-interpretation.

Reply: We do agree with the Reviewer’s previous comment re our relatively small number of male respondents potentially being a particularly motivated group (and thus not representative of the general population), we have deleted this sentence. 

Moreover:

5. The manuscript is unnecessarily long. Some paragraphs include quite expected or not so interesting information which adds very little to the main manuscript message and could be shortened or even removed to better focus on the main topic. For example lines 217-219 and relative paragraphs in the discussion.

Reply: We have made significant cuts to the manuscript, particularly in the Discussion, as per the reviewer’s request. Therefore, overall our word count has been reduced from 4,785 to 4,106.

REVIEWER #2 

1. The manuscript addresses an important an interesting problem, regarding parents/families acceptability of receiving predicted obesity risk information. It is well written, but needs to be more concise. The title specifically mentions acceptability of early childhood obesity prediction models, but parents have not been asked about obesity prediction at all. Neither have they addressed this question in their methods. 

Hence the main problem with the manuscript is the mismatch between what the parents are asked to comment on, and what the paper purports to be about. Whilst it is entirely appropriate that parents are not asked specifically about obesity, the authors need to address the mismatch in the title of the manuscript and the question actually posed to participants.

The manuscript refers many times to ‘obesity prediction’ and risk information, but nowhere is it the specific question posed to participants articulated in the manuscript. This is a major omission because the way the initial question is framed would likely affect the acceptability of the information.

There is one question in the appendix that refers to putting on too much weight by the time they start school. If this is the question that the authors refer to as ‘prediction information’ it needs to be in the main manuscript, but putting on too much weight ( weight trajectory) is quite different to ‘early childhood obesity prediction information’ referred to in the manuscript.

Additionally there is no example of the format the information might take. For example prediction information could be in the form of probability of childhood obesity, or probability of reaching above a heathy weight by a particular age etc etc. How will the authors convert from a prediction model to ‘chance of putting too much weight by the time they start school’????.

Reply: We thank the Reviewer for their comprehensive feedback regarding the specific wording of the central question in our manuscript. The Reviewer is correct in that we have based our findings on acceptance of prediction model information on our question regarding a child's chance of putting on too much weight by the time they start school. Additionally, a preamble at the start of the questionnaire stated: “There are ways of working out if a baby is likely to put on too much weight by the time they start school. We would like to hear from parents and caregivers whether it would be useful to know this information, and what help would be useful to parents if this was the case.” We have added this preamble to the start of Supplementary File 1.

It is true that the specific words “prediction model” were not used throughout our questionnaire. Māori, Pacific, and obesity researchers and early childhood organisations reviewed the wording of the questionnaire several times to ensure it was both culturally appropriate and understandable. We also received extensive feedback from our Māori and Pacific colleagues regarding the cultural inappropriateness of using phrases like “obesity”; “too much weight” was considered a more acceptable alternative. If such a prediction model were to be used in New Zealand this is likely to be the kind of language used to explain its purpose. Further, because the words “prediction model” were considered to not hold great meaning for a layperson without knowledge of statistical methods, we specifically chose to explain prediction models using simpler language. We have now added a statement to our Methods to explain which question we have used to measure acceptance of early childhood obesity prediction, as below.

Lines 104-114:

“An extended questionnaire was drafted and a refined version developed with input from Māori (indigenous people of New Zealand), Pacific, and other researchers, as well as relevant early childhood organisations to ensure cultural appropriateness, ease of understanding, and relevance. The specific question used to measure acceptance of early childhood obesity prediction was:

 “We are interested in how you as a parent or caregiver would like to be given information about your child's weight. For example, at your Well Child check in their first 6 months of life, the Well Child visitor could calculate if your baby has a greater chance of putting on too much weight by the time they start school. Would you like to know this information?” (Supplementary File 1).”

Results

2. It is unclear whether multivariable models were used for all question responses. The results appear to be simple summary statistics. If the multivariable model pertains to the main question, the model should be presented.

Reply: The previous version of the manuscript incorrectly referred to “models” in both the Methods and Results. In fact, only the responses for the question regarding acceptability of the prediction model information was analysed using a multivariable model. Summary results for this model are now presented in the manuscript, as well as full results in the newly created Supplementary Table 2. 

Lines 194 to 203:

“The results from the multivariable model are provided in Supplementary File 2. There were no differences between European, Māori, and Pacific respondents; the only distinct group were Asians (mean score 1.68), who were more accepting of the model information compared to European (2.05; p=0.003), Māori (2.11; p=0.002), and Pacific (2.03; p=0.042) respondents (Supplementary File 2).

Male respondents (mean score 1.83) were more accepting than females (2.11; p=0.005). In addition, grandparents were markedly more accepting (mean score 1.67) than parents (2.10; p=0.0002) and other caregivers (2.13; p=0.021) (Supplementary File 2). Respondents' acceptance of the information did not differ according to SES (1.91 vs more deprived 2.02; p=0.09) or level of education (university 1.94 vs high-school or lower 1.99; p=0.50).”

In addition, we have removed any references to multiple models from the manuscript.

Lines 167 to 171:

“Factors associated with acceptance of early childhood obesity were examined using a general linear model, including the following categorical predictors: sex, ethnicity (European, Māori, Pacific, and Asian), education level (complete/incomplete university qualification vs high-school or lower), caregiver type (parent, grandparent, or others), and SES (less deprived vs more deprived half).”

Line 194:

“The results from the multivariable model are provided in Supplementary File 2.”

We have also added a footnote to Table 4, detailing the statistical analysis used to compare the data presented in the table.

“The proportions of respondents within demographic characteristics were compared using chi-square tests.”

We have also clarified the two analyses carried out in the Methods section at Lines 167-174:

“Factors associated with acceptance of early childhood obesity were examined using a general linear model, including the following categorical predictors: sex, ethnicity (European, Māori, Pacific, and Asian), education level (complete/incomplete university qualification vs high-school or lower), caregiver type (parent, grandparent, or others), and SES (less deprived vs more deprived half). The proportions of respondents who provided their own and/or their child's height/length and weight were compared within demographic characteristics using chi-square tests. Data were analysed using SPSS v25 (IBM Corp, Armonk, USA). All tests were two-tailed, with significance level at p<0.05.”

3. Line 172 ‘like to know the prediction information’ – it appears they were not asked this at all.

Reply: Please see response to point 1. 

4. Line 209..’want to know the model’s information about their child’s weight’ – there is no question in the appendix that asks about information from a model.

Reply: As above, please see response to point 1. 

5. Line 227-239 – too much detail. If this is in the tables, please don’t repeat in the text.

Reply: The Reviewer makes a valid point, and the text has been modified to reduce repetition.

Lines 239-248:

“Approximately 60% (59.4%) of respondents “often” or “sometimes” had concerns about their own weight, and of these, 62.4% either “definitely” or “probably” wanted to know the prediction information on their child. There was no clear trend regarding differences between obese and non-obese respondents’ levels of interest in the prediction information according to their weight status, or that of their child’s (Table 3). However, respondents who provided their own weight and height were slightly more receptive to the prediction information (i.e. responding "definitely yes" or "probably yes") than those who did not (64.4% vs 56.8%, respectively; p=0.001) (Table 3). Of note, sociodemographic characteristics of those who provided their own or their child’s anthropometric data were markedly different to those who did not, with this information being more frequently provided by those who were university educated, from households with less deprivation, or of European ethnicity (Table 4).”

6. Figure 2 This is a nice figure, but the axis scales should be consistent, at least for the two different question types.

Reply: We have amended this figure so that the axis scales are now consistent across all questions. In addition, we have amended the axis scales of Figure 4 for the same reason.

Supplementary appendix page 2

7. Question beginning ‘We are interested….. calculate if your baby has a greater chance of putting on too much weight by the time they start school.’ – greater chance than what? Average? Should this have been high chance?

It is unclear whether this is the main question that has been interpreted as the prediction information

Reply: As previously explained, the Reviewer is correct in believing that this may have been the question interpreted as the prediction model acceptance question. While it might be argued that “high” may have been a better choice of words here, we do not believe use of the word “greater” was likely to hinder understanding of the question by respondents. 

8. The other major comment is the discussion is much too long, and at times veers off into a discussion of previous studies or aspects not relevant to the main theme of the paper. It needs to be much more concise. I would suggest reducing it to 25% of its current length.

Reply: We agree with the Reviewer, and a similar comment was also made by Reviewer 1. As a result, we have made significant reductions to the text as suggested, reducing our word count from 4,785 to 4,106.

Minor points

9. Abstract -Line 30 refers to infant, whilst results refers to child

Reply: We have made this edit to the Results section of the Abstract. 

Lines 39-41:

“Nearly two-thirds (62.1%) of respondents would “definitely” or “probably” want to hear if their infant was at risk of early childhood obesity, although “worried” (77.0%) and “upset” (53.0%) were the most frequently anticipated responses to such information.”

Introduction

10. Line 58 – preferable to whom?

Reply: We have further expanded this sentence to specify that we are referring to obesity prevention being preferable to treatment from a public health perspective (rather than a parental perspective).

Line 59-61:

“As long-term weight loss maintenance is difficult in children and adults, obesity prevention is preferable to treatment (6), from a public health perspective.”

11. Line 61 A number of prediction models – yet only two cited

Reply: Our citations (now at Line 64) refer to a commentary and review on the topic of early childhood obesity prediction models. The citations themselves refer to multiple models that have been developed worldwide.

12. Line 66 should this be overweight or obese?

Reply: The Reviewer is correct and we have edited this sentence for accuracy.

Lines 66-68:

“In addition, they have been developed for use prior to the age of 2 years, before an infant can be clinically considered overweight or obese (7).”

13. Line 69 – country context of the models should be mentioned

Reply: In response to the Reviewer’s feedback, we have added the country context of the models.

Lines 71-72:

“To date, two UK-based studies have tested use of such models with parents, one as a mobile phone application and the other as part of a feasibility study (7,8).”

14. Line 80 – and who? GP/nurse/childhood educator

Reply: We have now edited this sentence to add who were giving parents feedback about their children’s weight status.

Lines 83-85:

Although some studies have assessed parental views of receiving feedback regarding their child’s weight status from researchers or school-based weight screening programmes (11-14), only one UK-based study has explored parents’ views of prediction models for childhood obesity (15).”

15. Line 81 thoughts ….should be views

Reply: This change has been made.

Line 83:

“…only one UK-based study has explored parents’ views of prediction models…”

16. Line 122 and throughout- early childhood should be defined at least once. Does this mean under 5 years?

Reply: We have edited this sentence to define early childhood obesity in the context of our study.

Lines 136-138:

“Here, we focus on questions about caregivers’ perceptions of prediction of early childhood obesity (defined as obesity before a child begins school, which typically occurs at 5 years of age in New Zealand) and their acceptance of this information (Supplementary File 1).”

17. Line 142 replace BMI with BMI values

Reply: This change has been made.

Line 157: 

“Children's BMI values were converted into BMI z-scores...”

REVIEWER #3 

1. This paper is very interesting and well-written piece of work. I enjoyed reviewing it. I have a few minor comments that I feel may strengthen the paper.

Reply: We are grateful to the Reviewer for their positive feedback. 

Abstract:

2. Lines 34-35: I would remove this sentence (in this form) from the abstract. Rather say (in the results, perhaps) that there was representation across all ethnicities and levels of education

Reply: As suggested, we have removed this sentence from the Methods in the Abstract, and added the statement below to the Results (also in the Abstract).

Lines 38-39:

“Responses were received from caregivers of various ethnicities and levels of education.”

3. Lines 42-43: To what were males, Asians and grandparents more receptive? This sentence is a bit vague and needs %s to be more meaningful.

Reply: We have modified this sentence to reflect the extent to which males, Asians, and grandparents were more receptive to the prediction model information.

Lines 41-45:

“With lower mean scores reflecting higher levels of acceptance, grandparents (mean score = 1.67) were more receptive than parents (2.10; p=0.0002) and caregivers (2.13; p=0.021), while males (1.83) were more receptive than females (2.11; p=0.005), and Asian respondents (1.68) more receptive than those of European (2.05; p=0.003), Māori (2.11; p=0.002), or Pacific (2.03; p=0.042) ethnicities.“ 

Intro:

4. There is mention of different ethnicities in the abstract, but little mention of how this is relevant in the introduction. Is there data from NZ that suggests that children under 5 of differing ethnicities differ in terms of overweight/obesity? Are ethnically-different parents different as well (in terms of their own overweight/obesity, as well as education)?

Reply: There are marked differences in rates of childhood and adult obesity according to ethnicity and socioeconomic status in New Zealand. We have extended the final paragraph of the Introduction to explain that these differences are considerable in New Zealand.

Lines 87-91:

“While that study was a useful first step into understanding parental views of early childhood obesity prediction, it may have limited relevance for New Zealand’s diverse population, where obesity rates among children and adults vary considerably according to ethnicity and socioeconomic deprivation (2, 16). Of note, in 2015/16, 20.9% of Māori and 30.1% of Pacific 5-year-olds had obesity compared to 12.7% of Europeans (2). For Asian children, this figure was just 8.1% (2).” 

Methods:

5. It would be useful to add stats regarding internet use and access in NZ. This is available through the world bank (https://data.worldbank.org/indicator/IT.NET.USER.ZS?locations=NZ) and may help state your case for using a survey that is distributed through social media.

Reply: We thank the Reviewer for the very helpful suggestion, and have added the above citation and relevant data to our manuscript. 

Lines 117-118:

“Internet access is high in New Zealand, with over 90% of the population using the Internet at least once in a three-month period (20).”

6. Line 123-124: Did parents indicate which of their <5 child they chose? Is there potentially some bias in this?

Reply: We did not ask respondents to indicate which of their children <5 years they chose. We believe it is unlikely that there is bias in this. For example, even if a respondent chose to focus on a normal weight rather than overweight child, it is unlikely their responses would be affected, as they would still be the parent of an overweight child. 

Results:

7. Lines 220-221: The sentence that explains overweight and obesity could be reworded to be clearer, and the insertion of the number of children would be helpful. The sentence could read as follows: n=?(%) were classified as overweight/obese. Of these children, n=?(%) were obese. Otherwise, it may be easier to split the numbers (X were overweight, Y were obese).

Reply: We agree with the Reviewer that the previous wording was confusing, and have amended the text accordingly.

Lines 233-235:

“Among respondents who provided anthropometric information for their child and stated that their child's weight gain had been fine or insufficient (n=627), 59 (9.4%) had a child with overweight and 103 (25.8%) with obesity.”

Discussion:

8. Lines 291-294: This information would fit well in the introduction (as per previous comment)

Reply: We agree with the Reviewer and have moved this information to the Introduction (as outlined at point 4 above).

Strengths/Limitations:

9. Lines 395-397: The sentence starting with “Women living with overweight or obesity…” seems misplaced and doesn’t add value in this section. 

Reply: We thank the Reviewer for their feedback. The purpose of this sentence was to highlight how concerns about infant weight gain may differ according to maternal weight status. However, we agree that the information was misplaced in this section. Therefore, we have moved our reflection on the provision of anthropometric data by respondents to earlier in the Discussion at Lines 308-319:

“Interestingly, there appeared to be no differences in respondents’ interest in the prediction information according to their own or their child’s weight status (obese or non-obese). However, those who did not provide their own weight and height were apparently less accepting of the prediction information. There were clear demographic differences between those who did or did not provide either their own or their child’s anthropometric data, with those who are most likely to be affected by obesity (Māori and Pacific respondents, those with less than university education, and those of lower affluence), being the least likely to respond to these questions. In the UK, women living with overweight or obesity preferred larger infants, and did not express any concerns about health risks associated with childhood obesity (39). While there did not appear to be any differences in respondent's acceptance of the prediction information according to whether or not they provided their child’s weight and height in our study, this may simply be because they did not know this information (rather than intentionally not disclosing it).”

---

## [Editor Report · Decision Letter 1]

31 Oct 2019

Acceptability of early childhood obesity prediction models to New Zealand families

PONE-D-19-17554R1

Dear Dr. Butler,

We are pleased to inform you that your manuscript has been judged scientifically suitable for publication and will be formally accepted for publication once it complies with all outstanding technical requirements.

With kind regards,

David Meyre

Academic Editor

PLOS ONE
---

## [Editor Report · Acceptance letter]

19 Nov 2019

PONE-D-19-17554R1 

Acceptability of early childhood obesity prediction models to New Zealand families 

Dear Dr. Butler:

I am pleased to inform you that your manuscript has been deemed suitable for publication in PLOS ONE. Congratulations! Your manuscript is now with our production department. 

With kind regards,

on behalf of

Dr David Meyre 

Academic Editor

PLOS ONE